# The Return of Large Carnivores and Extensive Farming Systems: A Review of Stakeholders’ Perception at an EU Level

**DOI:** 10.3390/ani11061735

**Published:** 2021-06-10

**Authors:** Marcello Franchini, Mirco Corazzin, Stefano Bovolenta, Stefano Filacorda

**Affiliations:** Department of Agri-Food, Environmental and Animal Sciences, University of Udine, Via delle Scienze 206, 33100 Udine, Italy; mirco.corazzin@uniud.it (M.C.); stefano.bovolenta@uniud.it (S.B.); stefano.filacorda@uniud.it (S.F.)

**Keywords:** coexistence, European Union, human–carnivore conflict, livestock system, predator

## Abstract

**Simple Summary:**

Large carnivores and husbandry practices are important contributors to biodiversity integrity. However, conflicts that may arise when carnivores and livestock share the same areas may undermine both carnivore conservation and the maintenance of husbandry activities. Through the revision of the existing literature regarding human–carnivore conflict at a European Union (EU) level, our work aimed to assess stakeholders’ perception towards large carnivores (bear and wolf). The results showed that those categories (i.e., rural inhabitants and hunters), which were affected the most by the presence of carnivores were those who showed the most negative attitude. We showed that direct experience with carnivores has led the opinion of certain categories to become more negative. Furthermore, we did not find differences in terms of degree of tolerance by comparing areas in which carnivores and humans have coexisted for centuries and areas in which carnivores were extirpated. In the light of carnivore population dynamics, we recommend monitoring changes in attitudes over time to define appropriate solutions aimed at mitigating carnivore impacts.

**Abstract:**

Conflicts between large carnivores and human activities undermine both the maintenance of livestock practices as well as the conservation of carnivores across Europe. Because large carnivore management is driven by a common EU policy, the purpose of this research was to assess stakeholders’ perception towards bears and wolves at an EU level. We conducted a systematic search and subsequent analysis of 40 peer-reviewed studies collected from 1990 to September 2020 within Member States of the EU. Rural inhabitants and hunters exhibited the most negative attitude compared to urban inhabitants and conservationists, whose attitude was more positive. We showed that direct experience with predators as a consequence of ongoing re-colonization may have affected the degree of acceptance of certain categories and that the long-term coexistence between humans and carnivores does not necessarily imply increased tolerance. To encourage coexistence, we recommend monitoring changes in attitudes over time relative to carnivore population dynamics.

## 1. Introduction

Extensive grazing practices are less modified and more biodiverse than intensive livestock systems [1] and play a fundamental role in both the management and the conservation of areas of high natural value since these are important providers of ecosystem services (e.g., food, climate regulation, habitat and biodiversity maintenance, etc.) that contribute to human well-being [2,3]. Their importance in production, environmental, and social terms is recognized by European agricultural policies, which also provide direct payments for the public goods offered to society [4]. In the second half of the 20th century, drivers such as urban expansion [5], the shift towards tourism [6], agricultural intensification in the lowlands supported by institutional reforms (e.g., Common Agricultural Policy) [7], and land abandonment in less-favoured areas [8] markedly reshaped the traditional agricultural landscapes [7]. Traditional extensive livestock systems of Europe’s mountainous areas have been particularly affected [9] with subsequent abandonment, mainly of upland and less productive areas [10]. These re-naturalization phenomena have favoured the return of shrub and arboreal vegetation [11] and, consequently, of wildlife species including large carnivores [12]. In areas where human activities continue, wildlife and livestock activities overlap geographically, and such co-occurrence may lead either to positive or negative interactions [13,14]. At a global level, human–wildlife interactions arise for several reasons including progressive human advancement into wilderness areas [15], wildlife population range expansion [16], and wildlife recovery because of successful conservation plans [17]. Large carnivores are the most conflictual species that might exert a negative impact on human activities [18,19]. The existing conflict of interest between carnivore conservation and extensive grazing practices elicits strong emotional responses that may undermine both carnivore survival and the long-term maintenance of traditional husbandry practices [19,20,21,22,23,24]. Apex predators exert a key role in the maintenance of ecological balance as a consequence of trophic cascade effects [25,26]. Such top-down effects in fact play a major role in regulating ecosystem structure because of both direct (density-mediated) and indirect (behaviourally mediated) impacts on herbivores and other medium-sized carnivore species [25,26]. On the other hand, extensive livestock systems are multifunctional as they provide food and raw materials (e.g., water, fodder, wood, etc.) [27], support for human health through climate regulation, medical plants, and the prevention of soil erosion [28], as well as recreational or cultural activities [29]. Therefore, implementing effective mitigation measures in conflict hot-spots assumes remarkable importance in the preservation of carnivore populations while, on the other hand, fostering the maintenance of traditional husbandry practices.

### Theoretical Framework: The Perceptions towards Bears and Wolves

Brown bear (*Ursus arctos*) and grey wolf (*Canis lupus*) (hereafter carnivores) are controversial species that have returned to occupy part of their historical distribution range and are now legally protected in most European countries [30]. The recent return of such predators has evoked several emotions varying from admiration to a desire for their extirpation [31,32,33]. Several factors may be involved in the perception of carnivores. Firstly, sex, age, and education may play a key role in fostering positive attitudes. In general women, elderly people, people with a lower level of education and less knowledge of the target species show less tolerance [34,35,36]. Secondly, folklore referring to oral traditions, folk tales, culturally transmitted fear, distaste or love towards certain groups of animals may lead to important conservation issues as some species may survive to the detriment of others [37,38,39]. Thirdly, people living in rural areas are generally less tolerant than urban inhabitants [35,36,40,41,42], which in turn is linked to another factor that may drive people’s attitudes, i.e., direct experience with carnivores [34,36]. Urban interests are perceived as the dominating norm in society driving political processes and controlling policymaking processes [43]. Contrariwise, as far as political power is concerned, rural inhabitants are perceived to be at a lower level than urban ones [44]. This perception of political subordination is even more clear in relation to carnivore management and creates a situation in which rural people perceive that they are not considered, taken seriously, or given enough participation in the carnivore policy processes [45]. All these situations contribute to generating political alienation in terms of a general mistrust that rural inhabitants have towards actors and institutions of the political system [39,40]. All these situations may promote illegal killing of carnivores with common silence and appraisal by local stakeholders who see hunting violators as defenders of human safety [46,47]. On the basis of the information collected, the purpose of this study was thus to provide an initial comprehensive assessment of the level of acceptance of different stakeholder categories (with special focus dedicated to farmers and livestock owners) in relation to bear and wolf presence within the European Union (EU).

## 2. Materials and Methods

### 2.1. Literature Search

To answer these questions, from September to December 2020, we retrieved peer-reviewed English language scientific material published between 1990 and September 2020 addressing human–wolf/bear conflicts in EU countries. To do so, we used three comprehensive databases (Scopus, Web of Science, Pubmed). Only those countries that have a permanent and reproductive presence of at least one species or are affected by the occasional presence of a given species (e.g., animals in dispersal) without reproduction [12,48,49,50] (i.e., Austria, Belgium, Bulgaria, Croatia, Czech Republic, Denmark, Estonia, Finland, France, Germany, Greece, Hungary, Italy, Latvia, Lithuania, Netherlands, Poland, Portugal, Romania, Slovakia, Slovenia, Spain and Sweden) were included in the analysis. Likewise, those countries in which the presence of such predators (neither stable nor sporadic) has not been reported in recent years (i.e., Cyprus, Ireland, Luxemburg, Malta) [12,48,49,50] were excluded. The United Kingdom, despite being part of the EU up to 2019, was excluded from the analysis as these carnivores were eradicated there well before 1990 [12]. The literature search was carried out using the following research string: *conflict* OR attack* OR predation OR damage* AND management OR retaliation OR kill* OR poach* OR mortality OR cull* OR control OR mitigation OR prevention OR attitude OR perception OR compensation AND wild* OR predator OR carnivore* OR bear* OR wolf OR wolves AND zootechny* OR husbandry OR transhumance OR extensive OR graze* OR rural OR rangeland OR farm* OR pasture* OR livestock OR cattle OR sheep OR goat**. The inclusion of “*” was made to include all the possible variations of the word considered (e.g., *cull** ⟶ *cull, culling, culled*). With this screening, we were left with 5040 publications (1217 from Scopus; 2959 from Web of Science; 864 from Pubmed). We then manually screened the remaining publications to identify studies that dealt with depredation on livestock species (cattle, sheep, goats) to evaluate stakeholders’ attitude towards both wolves and bears. In those studies (e.g., Swedish ones) in which the attitude towards a larger range of predators (e.g., bear, wolf, lynx, wolverine) was evaluated, we considered only data referring to bears and wolves. The same criteria were used for the different husbandry species (e.g., sheep, reindeer) mentioned in the research. In such cases, only information related to the target livestock species was included in the research (i.e., sheep). In those studies in which stakeholders’ attitude was evaluated in more countries all belonging to the EU (e.g., Italy and Greece), we discerned the information reported in both areas (i.e., attitudes for Italy and attitudes for Greece). On the contrary, in those studies in which the attitude of the different stakeholder categories was evaluated in more countries not all belonging to the EU (e.g., Norway and Sweden), we considered only the information reported in EU countries (i.e., Sweden). After removal of duplicates, articles that included either livestock or carnivore species not pertinent to the present research and articles whose topic was out of our scope (e.g., those dealing with human–carnivore conflicts but not referring to stakeholder perceptions), the potential sample was reduced to 40 pieces of scientific research (Table 1; Figure 1). Countries included in the review are shown in Figure 2.

### 2.2. Literature Content Analysis

To assess the attitude towards bears and wolves, we performed a meta-analysis using information obtained by standardized questionnaires, a 3- or 5-point Likert-scale, or by using the information reported in each scientific paper (e.g., livestock owners stated that they have low tolerance towards large carnivores or are in favour of the local eradication of the species). Attitudes were standardized as negative, neutral or positive based on the respective Likert-scaled points used in each questionnaire. When considering the 5-point Likert-scale, attitudes were considered negative if the answer included points 1 and 2, neutral for point 3, and positive for points 4 and 5. The same criteria were used for the 3-point Likert-scale (i.e., 1 = negative; 2 = neutral; 3 = positive). When the attitude was reported as “nuanced” or “variable”, depending on specific situations (e.g., direct experiences with the target species) we considered it as a neutral in the analysis. When the attitude varied between neutral and lower, a negative attitude was considered and vice-versa for attitudes varying from neutral to positive. The sampled groups were classified as follows: urban inhabitants, rural inhabitants (mainly farmers and livestock owners), hunters, general public (pet dog owners, guesthouse owners, local educators, berry and mushroom pickers, hikers, fishers, hotel employers, teachers, housewives, pensioners, employees, students), and conservationists (scientists, environmentalists, nature conservationists, Non-governmental organizations, park members, foresters). As far as the general public and/or rural inhabitants were concerned, in some cases, some of them were also hunters. Therefore, the attitude reported was divided between the two categories (e.g., if rural people who were also hunters expressed a negative attitude, in the analysis a negative attitude was considered for both rural people and hunters).

To calculate the attitude of each category, the following index was used:*A_i_* = ((*x*1 * *k*1) + (*x*2 * *k*2) + (*x*3 * *k*3))/*n*
where:*A_i_* = attitude of the *i*-th category*x*1 = number of cases in which a negative attitude was mentioned*k*1 = 1 (value arbitrarily defined for a negative attitude)*x*2 = number of cases in which a neutral attitude was mentioned*k*2 = 2 (value arbitrarily defined for a neutral attitude)*x*3 = number of cases in which a positive attitude was mentioned*k*3 = 3 (value arbitrarily defined for a positive attitude)*n* = *x*1 + *x*2 + *x*3

The choice to use such an index was taken in order to obtain a comprehensive measure of the attitude of each stakeholder category, taking into consideration the number of times in which a specific attitude was reported by each category and the overall number of attitudes (positive, neutral or negative) reported.

To compare the attitude between areas in which coexistence had ever persisted with those in which carnivores had been eradicated, because of the limited information available for the other stakeholders, we referred only to those categories that showed the most negative attitude, i.e., rural inhabitants and hunters (see *Results*), and included only studies in which information regarding the status and distribution of both bears and wolves across the study area was provided. To answer this question, the analysis was conducted on a small scale (i.e., considering only the small area/s in which the study was carried out) and considering each case as independent. For instance, for Italy, we included two comparative studies: one carried out in the central Alps (where bears were almost totally eradicated) [60] and another in the central Apennines (where humans and bears have coexisted for centuries) [61] and both were considered as independent cases. This choice was made because in some countries (such as Italy, for example), carnivore eradication took place at a local scale and not throughout the country. Therefore, if we had considered the whole country as an area in which carnivores had persisted and/or were totally eradicated, we would have made a conceptual mistake.

### 2.3. Statistical Analysis

Statistical analyses were performed using R Software (version 4.0), and the alpha value was set at 0.05. To test the difference in terms of attitude between categories, Fisher’s test [78] was applied, using the number of times (reported as *n* in the *Results* Section) in which a certain attitude was reported by the i-*th* category as an observed frequency. To best cope with the reduced sample size, comparison between categories (e.g., positive vs. neutral, positive vs. negative, neutral vs. negative) was done using the pairwise nominal independence function through the R package *rcompanion* [79]. Because of the diversity in terms of publications among years and considering the time interval in which such publications were produced, i.e., 14 years (2003, 2004, 2006, 2008, 2009, 2011, 2012, 2013, 2014, 2015, 2016, 2017, 2019, 2020), we performed a subdivision between publications published between 2003 and 2012 (hereafter, the first period, *n* = 13) and those from 2013 to 2020 (hereafter, the second period, *n* = 27). This was done to realize a proper comparison between two-time periods of seven years each. Fisher’s test was further used to compare attitude variations between periods and between areas in which carnivores were eradicated (before then starting to recolonize their former range) and those in which rural inhabitants/hunters and carnivores had ever coexisted. Comparisons in terms of the attitude index between categories were realized using the non-parametric Kruskal–Wallis *H* test [80]. The same test was used to compare attitudes between periods and both the coexistence and non-coexistence areas.

## 3. Results

### 3.1. General Attitude towards Carnivores

From the comparison in terms of stakeholders’ attitude towards carnivores, we found that urban inhabitants showed no differences between negative (*n* = 2, 22%) and neutral attitudes (*n* = 1, 11%), while a significant difference was obtained between negative and positive attitudes (*n* = 9, 67%) (F-test, *p* = 0.02), as well as between a neutral and positive attitude. Therefore, as reported by the attitude index (A_urban_ = 2.44), urban inhabitants showed, in general, a more positive attitude. As far as rural inhabitants were concerned, a significant difference was found between a negative (*n* = 27, 73%) and neutral attitude (*n* = 6, 16%) (F-test, *p* < 0.001) as well as between a negative and a positive attitude (*n* = 4, 11%) (F-test, *p* < 0.001) and between a neutral and a positive attitude (F-test, *p* < 0.001). Hence, as confirmed by the attitude index (A_rural_ = 1.38), rural inhabitants revealed a more negative attitude. A similar trend was observed for hunters. A significant difference was noted between a negative (*n* = 14, 78%) and a neutral attitude (*n* = 1, 6%), between negative and positive attitudes (*n* = 3, 17%) (F-test, *p* < 0.001), and between a neutral and a positive attitude (F-test, *p* < 0.001). Nevertheless, hunters (A_hunters_ = 1.39) exhibited a slightly less negative attitude than rural inhabitants. As far as the general public was concerned, Fisher’s test reported no significant difference among attitudes (n_negative_ = 8, 33%; n_neutral_ = 5, 21%; n_positive_ = 11, 47%) (F-test, *p* = 0.11). This means that, as confirmed by the attitude index (A_public_ = 2.13), the general public generally exhibited a neutral attitude. Conservationists was the category that revealed the strongest positive attitude (A_conservationists_ = 2.64). A significant difference was in fact obtained between a negative (*n* = 1, 7%) and a neutral attitude (*n* = 3, 21%) (F-test, *p* < 0.001), between a negative and positive attitude (*n* = 10, 71%), and between a neutral and a positive one (F-test, *p* < 0.001). No significant difference was found comparing the attitude indexes between categories (KW-test, *χ2* = 4, *p* = 0.41). The frequency distribution of the answers reported is shown in Figure 3.

### 3.2. Attitude Comparison between Periods

The attitude of urban inhabitants did not change markedly from the first (n_negative_ = 2, 40%; n_neutral_ = 0, 0%; n_positive_ = 3, 60%) to second period (n_negative_ = 1, 25%; n_neutral_ = 1, 25%; n_positive_ = 2, 50%). Despite showing a neutral attitude, the attitude index during the second period (A_urban_ = 2.25) was slightly higher than that shown during the first (A_urban_ = 2.20) firstly because, in the initial period, we did not encounter cases in which a neutral attitude was reported (n_neutral_ = 0) and, secondly, following the formula above, in the first period the number of cases in which attitudes were reported was higher (*n* = 5) than in the second (*n* = 4). Furthermore, no significant difference was found in terms of attitude index between periods (KW-test, *χ2* = 1, *p* = 0.32).

The attitude of rural inhabitants remained negative (F-test, *p* < 0.001) in both the first (n_negative_ = 9, 70%; n_neutral_ = 2, 15%; n_positive_ = 2, 15%) and second periods (n_negative_ = 21, 84%; n_neutral_ = 3, 12%; n_positive_ = 1, 4%), but in the second (A_rural_ = 1.20), this was even more negative than that in the first (A_rural_ = 1.46). Nevertheless, no significant difference was found between periods (KW-test, *χ2* = 1, *p* = 0.32).

The attitude of hunters changed from neutral (F-test, *p* = 1.00) in the initial period (n_negative_ = 3, 50%; n_neutral_ = 0, 0%; n_positive_ = 3, 50%) to negative (F-test, *p* < 0.001) in the second, where only negative responses were recorded (n_negative_ = 11, 100%; n_neutral_ = 0, 0%; n_positive_ = 0, 0%). This trend was also confirmed by the attitude index values recorded in both the first (A_hunters_ = 2.00) and second periods (A_hunters_ = 1.00). However, no significant difference was found in terms of the attitude index between periods (KW-test, *χ2* = 1, *p* = 0.32).

The attitude of the general public changed from positive (F-test, *p* < 0.01) in the first period (n_negative_ = 3, 21%; n_neutral_ = 3, 21%; n_positive_ = 8, 57%) to neutral (F-test, *p* = 0.98) in the second (n_negative_ = 4, 29%; n_neutral_ = 5, 36%; n_positive_ = 5, 36%) as confirmed by the respective attitude index values (A_public_ = 2.36 and A_public_ = 2.07 for the first and second periods, respectively). No significant difference was found in terms of the attitude index between periods (KW-test, *χ2* = 1, *p* = 0.32).

The attitude of conservationists remained positive (F-test, *p* < 0.001) in both the first (n_negative_ = 0, 0%; n_neutral_ = 0, 0%; n_positive_ = 4, 100%) and second period (n_negative_ = 1, 11%; n_neutral_ = 2, 22%; n_positive_ = 6, 67%) even though in the first period this was totally positive (A_conservationists_ = 3.00) compared to the second, in which negative (*n* = 1) and neutral (*n* = 2) attitudes were reported (A_conservationists_ = 2.56). No significant difference was found in terms of the attitude index between periods (KW-test, *χ2* = 1, *p* = 0.32).

The comparison in terms of frequency distribution of the responses obtained between periods is shown in Figure 4a,b.

### 3.3. A Comparison of Rural Inhabitants’ and Hunters’ Attitudes between Coexistence and Non-Coexistence Areas

Comparing the attitude of rural inhabitants between areas in which humans and carnivores have always coexisted and the areas in which these carnivores were eradicated, we observed that the attitude was significantly negative in both coexistence (n_negative_ = 2, 67%; n_neutral_ = 1, 33%; n_positive_ = 0, 0%) and non-coexistence areas (n_negative_ = 12, 86%; n_neutral_ = 1, 7%; n_positive_ = 1, 7%). However, in areas where carnivores had been eradicated, this was even more negative (F-test, *p* < 0.001, A_rural_ = 1.21) than in areas where carnivores and humans have coexisted for centuries (F-test, *p* < 0.001, A_rural_ = 1.33). Nevertheless, we did not find a significant difference in terms of attitude indexes between areas (KW-test, *χ2* = 1, *p* = 0.32).

As far as hunters are concerned, the attitude was significantly negative in both coexistence (n_negative_ = 1, 100%; n_neutral_ = 0, 0%; n_positive_ = 0, 0%) and non-coexistence areas (n_negative_ = 7, 70%; n_neutral_ = 0, 0%; n_positive_ = 3, 30%). Nevertheless, even in this case, in areas where carnivores had been extirpated, the attitude was even more negative (F-test, *p* < 0.001, A_hunters_ = 1.60) than in areas where carnivores and humans have always coexisted (F-test, *p* = 0.02, A_hunters_ = 1.00). However, it is important to specify that the lower attitude index value in the coexistence area is linked to the only negative response obtained. Furthermore, even in this case, no significant difference was found in terms of the attitude index between areas (KW-test, *χ2* = 1, *p* = 0.32).

The comparison in terms of the frequency distribution of the responses obtained between areas is shown in Figure 5a,b.

## 4. Discussion

Our study represents one of the first attempts to provide a comprehensive understanding of the attitudes of various stakeholder categories towards bears and wolves at an EU level. From our study, we found that both hunters’ and rural inhabitants’ attitudes were strongly negative, while urban inhabitants and conservationists were more tolerant. The general public was the only category showing a neutral attitude.

Attitude towards carnivores may vary according to different cultural, socio–economic, and/or political circumstances. This in turn can be linked to socio–economical parameters, history, and wildlife management policies [57,81,82,83,84]. Factors such as age [85], sex [86], educational level [87], and direct experiences with target predator species [57,63] seem to exert a key role in driving people’s attitudes. For instance, Stauder et al. (2020) [63] showed that urban people living in areas where wolves were absent were more tolerant toward carnivores. By the same token, Piédallu et al. (2016) [57] showed that people living in no-bear areas showed more tolerance than those living in areas with bears. Rural inhabitants, especially farmers and livestock owners, represent the most affected category since the carnivores’ return is linked to increased clashes with extensive grazing practices, potentially generating conflicts amongst a range of categories. Indeed, the perceived asymmetry in terms of political power between urban and rural areas may even lead to the development of conflicts between urban and rural inhabitants as the latter feel excluded from the political system [41,42,44]. In the case of policy towards carnivores, livestock owners may view the effective (or perceived) reintroduction of carnivores as a sort of political oppression by urban groups [88]. Moreover, they perceive the presence of predators as a negative factor, potentially limiting their day-to-day activities [89]. Opposing the return of carnivores is then often seen as a necessity, both in the sense of maintaining traditional rural practices and defending their political autonomy in the face of urban interests [88].

We found that the attitude of the categories involved changed between periods (2003–2012 and 2013–2020), except for the general public whose attitudes remained neutral and conservationists whose attitudes remained positive. Indeed, the latter perceived the return of such predators as a positive point to re-establish and maintain ecological balance and to prevent the disruption of ecological systems [25,26]. The attitude of rural inhabitants remained negative in both periods, becoming even more negative during the second, while hunters’ attitude changed from neutral in the first period to negative in the second one. Urban people showed a positive attitude during the first period changing to neutral during the second. We interpreted such changes as being that urban residents find it easier to support the large carnivore’s return and conservation in areas where they had been eradicated because they did not experience interactions with them [57,63]. Indeed, these results are in line with findings presented by Dressel et al. (2015) [33], who compared the attitude of peoples towards both bears and wolves in Europe from 1976 to 2012.

As expected, the attitudes of rural inhabitants remained negative in both periods as they were more involved in the conflicts. Nevertheless, even among livestock owners, we found a relationship between tolerance and direct experiences with carnivores. Livestock owners who had experienced damage from carnivore attacks were more inclined to have a negative attitude than those that suffered little or no damage. Hunters’ attitude shifted from neutral to negative as they perceived the return of carnivores as a potential threat for hunting dogs. Moreover, they may see predators as competitors for large game species [54]. Of particular interest are the results obtained in terms of attitude comparisons between periods as far as conservationists are concerned. Contrary to the first period in which only positive attitudes were reported, during the second one, one negative [67] and two neutral attitudes [22,59] were registered. Niedziałkowski and Putkowska-Smoter (2020) [67] stated that some foresters benefitted from organising wolf hunts for Polish and international hunters, while others believe in and recognize the ecological value of wolves within the ecosystem (i.e., through limiting ungulate densities, they indirectly have a positive impact on forest plantations). However, at the local and regional levels, foresters did not particularly endorse wolf protection and sometimes outright criticized conservation initiatives. Anthony and Tarr (2019) [59] declared that some park members perceived the presence of wolves as positive since they reduce the number of damaging species such as wild boar, beyond removing weak and/or sick animals. On the other hand, others believe that all wolves should be killed or confined to zoos. Gosling et al. (2019) [22] found that foresters exhibited a neutral position, but such results may have been influenced by the fact that 35% of them were hunters and 32% held livestock. Despite a very small data set, these findings are interesting as they may suggest that, in line with carnivore recolonization, conservationists become aware that coexistence between people and predators may be impractical in some areas. Thus, they have probably started to change their exclusively conservationist position towards a conservationist and management one. However, as stated before, because of the small sample size, further research should be implemented to provide stronger inferences.

Because of the poor quality of the information available, comparisons in terms of attitudes between areas where humans and carnivores have always coexisted with those in which carnivores have been eradicated were carried out only with reference to rural inhabitants and hunters. In both cases, we observed a negative attitude in both areas, which was even more negative in recolonization zones. This result suggested that where carnivores and humans coexist, conflict occurs because of direct or indirect interactions. Thus, the attitude is generally negative. Moreover, in areas where people are no longer accustomed to coexisting with predators, they may perceive the return of carnivores as a sort of limit for either freedom [89] and/or livestock activities [88]. However, because of the very small number of studies reported, this result should be interpreted cautiously, and further research should be carried out.

From our research, we were unable to provide a broader evaluation in terms of degree of tolerance towards bears and wolves, as few studies (*n* = 3) reported discriminatary attitudes between the two species. Pohja-Mykrä (2016) [47] showed that hunters and livestock owners considered the wolf as the most problematic species. The same results were reported by Pohja-Mykrä (2017) [55] and Mykrä et al. (2017) [54], i.e., hunters and rural inhabitants showed a less positive attitude towards wolves than that shown towards bears.

## 5. Research Limits

We are aware that our research presents some limitations. Firstly, we focused only on peer-reviewed English literature, thus excluding the grey literature, which might have provided additional information. We decided to focus only on peer-reviewed studies because the scientific contribution represents an important focus of this work. Despite this, we believe that because of the nature of the topic, this exclusion criterion did not eliminate a large number of highly valuable studies. However, we recognize that the grey literature may represent a valuable source of information when more valuable peer-reviewed research is limited. Secondly, the attitude index value of each category involved was strongly affected by the number of studies in which such attitudes were obtained. Finally, marked differences emerged in terms of the number of studies published between the first (*n* = 13) and second period (*n* = 27) and between coexistence (*n* = 4) and non-coexistence areas (*n* = 16), which may have affected the results obtained.

## 6. Conclusions

The return of large carnivores in areas in which they were previously extirpated evoked different feelings among the different involved stakeholder categories. This not only refers to the European context but even includes other countries (e.g., India) in which human density has reached remarkable levels. Consequently, in such areas, interactions or conflicts between wild species (especially carnivores) and human activities may become very intense. Therefore, the synergistic participation of research institutions and local authorities should be implemented to find the most adequate solutions aimed at promoting coexistence in the long-term.

Our findings support the idea that the return of both bears and wolves is challenging for conservation, as interactions with such predators may alter people attitudes, particularly those of the most affected categories. Long-lasting coexistence does not necessarily imply that people are more willing to accept carnivores. Conservationists should thus continuously monitor public attitudes, as opinions change when carnivore populations become established. Moreover, changes in these carnivores’ demographic parameters need to be carefully taken into consideration by carnivore-policy makers to draw up effective management actions with the aim of minimising livestock killings. Because of the existence of a common EU policy, transboundary cooperation may help in the design of shared and effective mitigation strategies.

In the near future, research institutions and management authorities will have to cooperate to solve a growing variety of human–carnivore conflicts, especially in those contexts in which political, social, and ecological conditions are changing. Therefore, communicating effectively with the public to promote large carnivore conservation and the maintenance of husbandry practices takes on a notable importance. We recognize the utility of extensive grazing practices in terms of the ecosystem services provided, especially at an EU level, and we are aware that if biodiversity eradication to promote the expansion of rural activities is impractical and dangerous, on the other hand, conservationists should change their views regarding the separation between rural practices and nature, throughout accepting that livestock practices are part of the ecosystem. Therefore, coexistence should be promoted and communicated, not only focusing on the importance of wildlife preservation, but also by highlighting the needs of livestock grazing practices of great economic, traditional, and ecological values.

Supporting Information: The list of keywords used for the literature search is provided in the *Methods* Section while the references used in the meta-analysis have been embedded within the *References* Section along with those that were merely cited. However, to make the literature search easier, the list of publications used for the meta-analysis is summarized in Table 1. The authors are solely responsible for the content and functionality of these materials. All further queries should be directed to the corresponding author.

## Figures and Tables

**Figure 1 animals-11-01735-f001:**
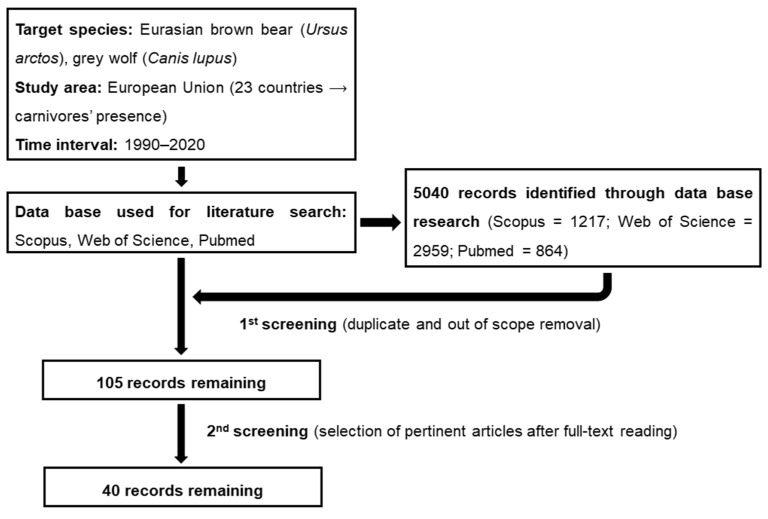
Overview of the criteria used for manuscript selection and dataset creation.

**Figure 2 animals-11-01735-f002:**
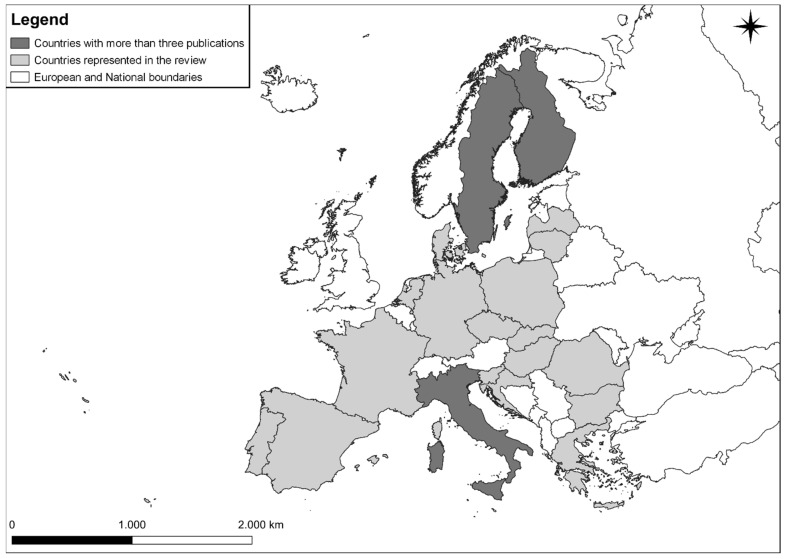
Countries represented in the review referring to the local perception of the involved categories (urban inhabitants, rural inhabitants, hunters, general public, conservationists). In Italy, we found five papers focused on the evaluation of stakeholders’ perception towards carnivores. In Sweden, ten scientific articles focused on stakeholders’ perception. As far as Finland is concerned, only four articles focusing on the attitude of stakeholders were found.

**Figure 3 animals-11-01735-f003:**
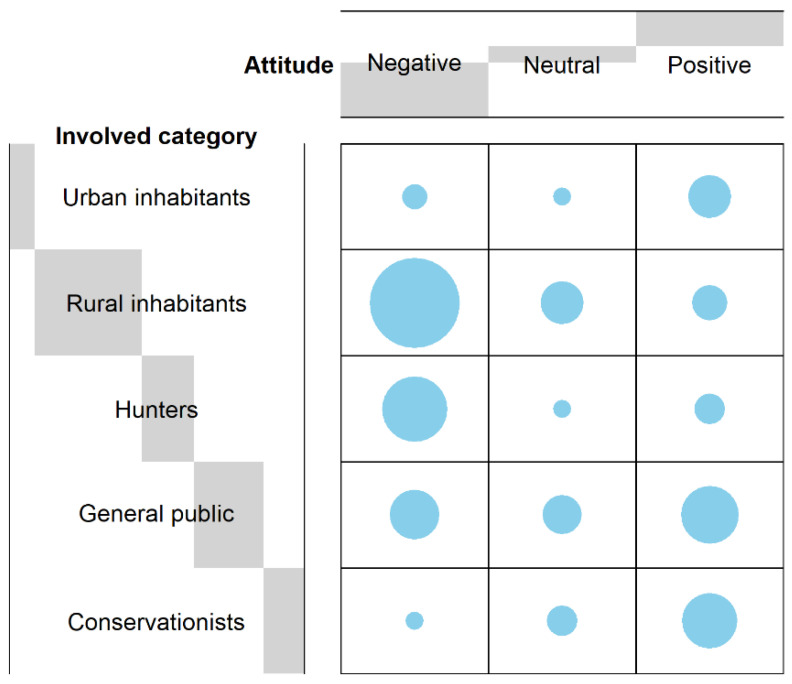
Contingency table showing the distribution frequency as far as the attitude towards carnivores of each category involved is concerned, i.e., urban inhabitants, rural inhabitants (mainly farmers and livestock owners), hunters, general public, conservationists. The size of the grey bars depends on the number of responses obtained by each stakeholder category for each attitude. For instance, from the above figure we see that the grey bar for rural inhabitants is the largest. This is because the highest number of responses were obtained (negative = 27, neutral = 6, positive = 4). On the contrary, the grey bar for urban inhabitants is the smallest as the lowest number of responses were obtained (negative = 2, neutral = 1, positive = 6). As far as the attitude is concerned (i.e., negative, neutral, positive), the criterion is the same. The negative attitude is that one which showed the larger grey bar as was that one mentioned the most by all stakeholders (urban inhabitants = 2, rural inhabitants = 27, hunters = 14, general public = 8, conservationists = 1). Contrarywise, the neutral attitude showed the smallest grey bar as the lesser mentioned by the stakeholders (urban inhabitants = 1, rural inhabitants = 6, hunters = 1, general public = 5, conservationists = 3). Reference list: [22,23,34,35,36,37,38,39,40,41,42,46,47,51,52,53,54,55,56,57,58,59,60,61,62,63,64,65,66,67,68,69,70,71,72,73,74,75,76,77].

**Figure 4 animals-11-01735-f004:**
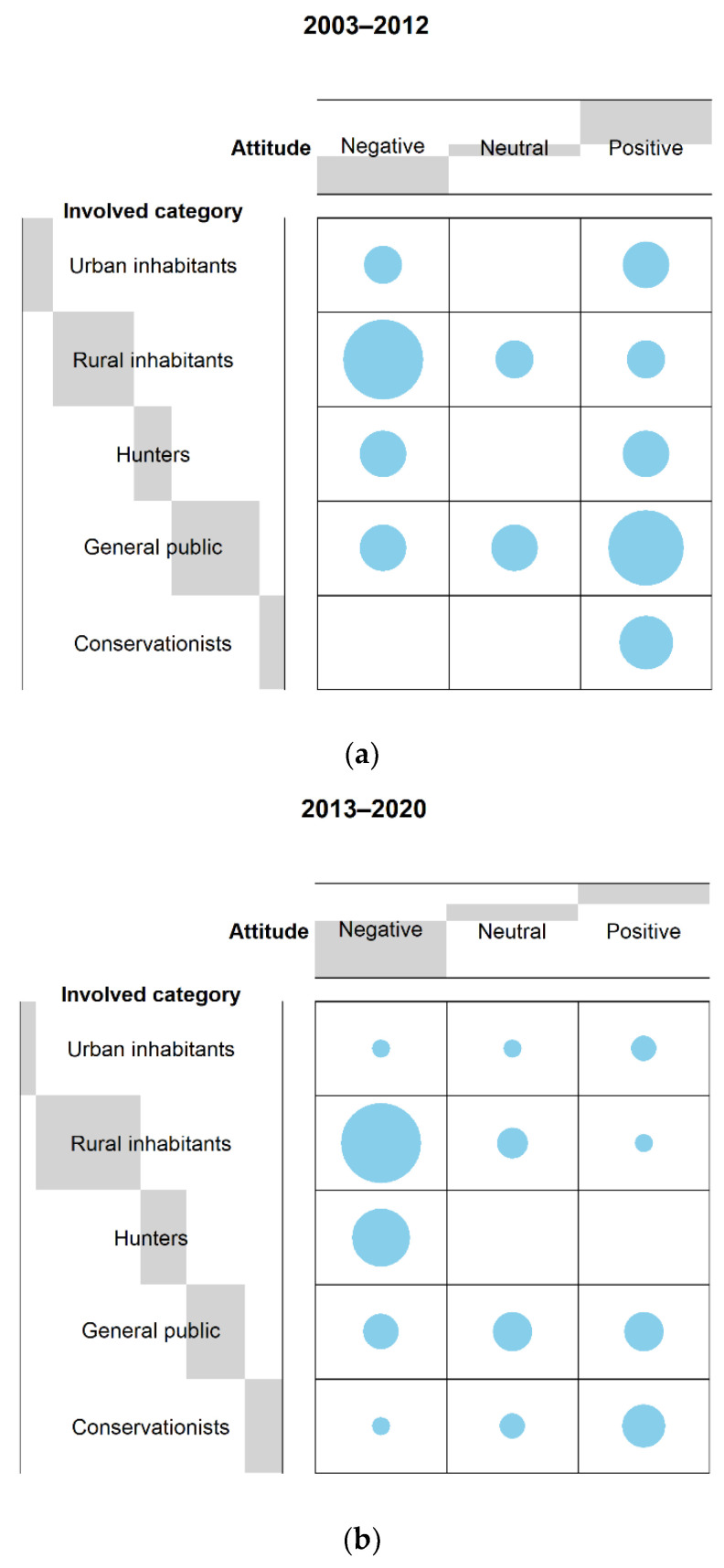
(**a**). Contingency table showing the distribution frequency regarding the attitude towards carnivores of each category involved in the first period (2003–2012). The size of the grey bars depends on the number of responses obtained by each stakeholder category for each attitude (refer to Figure 3 caption for a more detailed explanation). For the reference list divided by periods, refer to Table 1. (**b**). Contingency table showing the distribution frequency regarding the attitude towards carnivores of each category involved in the second period (2013–2020). The size of the grey bars depends on the number of responses obtained by each stakeholder category for each attitude (refer to Figure 3 caption for a more detailed explanation). For the reference list divided by periods, refer to Table 1.

**Figure 5 animals-11-01735-f005:**
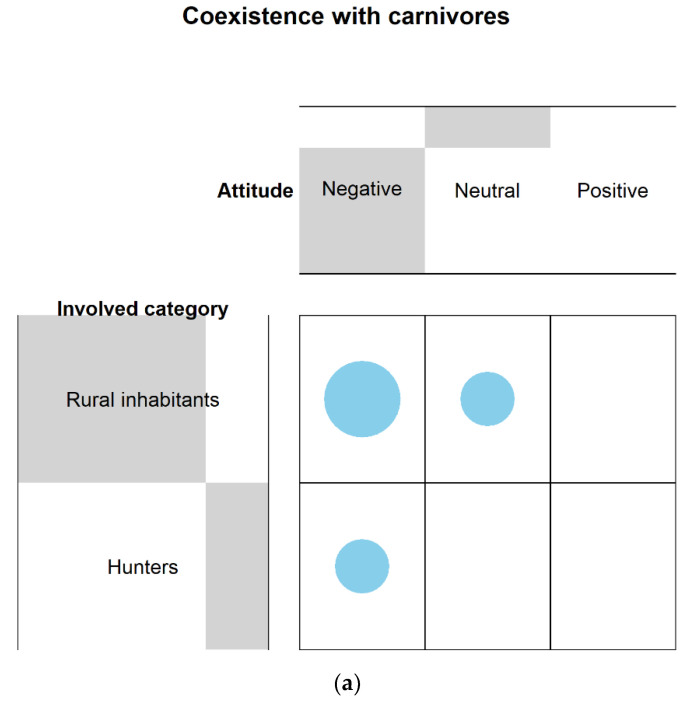
(**a**). Contingency table showing the distribution frequency regarding the attitude towards carnivores of rural inhabitants (mainly farmers and livestock owners) and hunters in areas where humans and carnivores have always coexisted. The size of the grey bars depends on the number of responses obtained by each stakeholder category for each attitude (refer to Figure 3 caption for a more detailed explanation). Reference list: [36,61,69,70]. (**b**) Contingency table showing the distribution frequency regarding the attitude towards carnivores of rural inhabitants (mainly farmers and livestock owners) and hunters in areas where carnivores have been eradicated. The size of the grey bars depends on the number of responses obtained by each stakeholder category for each attitude (refer to Figure 3 caption for a more detailed explanation). Reference list: [34,39,40,41,42,52,53,54,57,59,60,66,68,72,73,74].

**Table 1 animals-11-01735-t001:** List of publications used in the meta-analysis divided by period and country/ies.

Country/ies	Period	
	2003–2012	2013–2020
Czech Republic	-	[51,52]
Denmark	-	[53]
Finland	-	[46,47,54,55]
France	[56]	[57]
Germany	-	[58]
Hungary	-	[59]
Italy	-	[60,61,62,63]
Italy, Greece	[64]	-
Latvia, Lithuania, Bulgaria, Turkey	-	[35]
Lithuania	[40]	-
Netherlands	-	[34]
Norway, France	[65]	-
Norway, Sweden	-	[39,66]
Poland	-	[22,67]
Portugal	[68]	[36]
Romania	-	[69,70]
Slovakia	[71]	-
Slovakia, Romania, Croatia	-	[38]
Slovenia	[72]	-
Spain, Portugal	[37]	-
Sweden	[73,74,75,76,77]	[23,41,42]

## Data Availability

Not applicable.

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
