# Peer review of "The Return of Large Carnivores and Extensive Farming Systems: A Review of Stakeholders’ Perception at an EU Level"

_animals, 2021, doi:10.3390/ani11061735_

Round 1

Reviewer 1 Report

The authors conducted a systematic review analysing studies published from 1990 to 2020 on the topic of carnivore mitigation strategies and the perception of different stakeholder categories. They report on trends and attitude differences among European countries. The manuscript is well written, but more information needs to be included and the discussion section extended. Moreover, the analytical approach should be reconsidered.

Specific comments:

Citation (Agri-Food, D.o.;…) needs to be corrected to Franchini, M…. – check the journal’s guidelines for authors.

L22: replace “undermines” with “undermine”

L26: replace “predations” with “predation”

L27: replace “State members” with “Member States”

L129-157: It is important to specify the date on which the systematic search was conducted.

L156-157: These are not properly described exclusion and inclusion criteria that would allow for replication of your study. Please provide a specific description of how you proceeded.  

L163-164 and Figure 2: It is not clear how figure 2 represents the 70 studies included in the review. Consider adding another diagram or table to make clear which studies were included.

L197-205: Can you cite a reference that used the same methodology or described the scientific approach behind this methodology?

L218-220: This statement needs to be supported by a reference.

L244-245 and L275-276: These numbers (n=2, n=1) are too low to conduct any meaningful analysis! L469: replace “need” with “needs”

L439-505: The discussion section needs to be extended to discuss also studies conducted outside of Europe, on different animal species, on different mitigation approaches, in order to provide a more general context for your study.

L514: “limited” might be a better word here than “poor”

L515-518: Indeed, I agree. That is why the statistical approach used here is incorrect.

L555: You should provide the table that you created from the studies identified in the study so that one could check that your classification and assessment were correct.

Reviewer 2 Report

Summary: Dr Franchini and colleagues authored the manuscript entitled “The return of large carnivores and extensive farming systems: a review of stakeholders' perception and mitigation strategies to reduce livestock predation events at an EU level” where they presented the results of a literature data analysis on the attitudes and perceptions of some stakeholders towards large carnivores in various European countries. The authors searched for peer reviewed articles and analyzed the contents of 70 papers that contained information about stakeholder's attitudes towards wolf and/or bears. The analysis was made comparing perceptions reported by the most affected stakeholders (hunters, livestock owners and farmers –assumed to represent rural inhabitants-, the general public, conservationists, and urban inhabitants). An assessment is also made on the existence of scientific evidence that mitigation measures are effective for reducing livestock predations.

The results highlight some significant difference in attitudes at single group level, with the rural inhabitants showing more negative and the conservationists and general public showing more positive attitudes. No significant difference is detected when comparing all groups. A comparison was also made between two time periods: 2003-2012 vs 2013 -2020, which shows that the general public and conservationists hold less positive attitudes in the more recent years than the previous ones.

General: The results confirm what already emerged in other studies, e.g., the rural-urban divide (see among others, Eriksson, 2016, 2017; Heberlein and Ericsson, 2005; Dressel et al., 2015), the issue of political representativeness and social alienation and the lack of scientific evidence on the effectiveness of damage prevention measures. The paper does not add much to the existing literature, except for a focus on the European literature, as an assessment of evidence to adequately evaluate the effectiveness of damage prevention methods was published by Eklund et al. in 2017. Those authors performed a review of peer reviewed articles published worldwide from 1990 to 2016, while Franchini et al. are presenting results from a review of peer reviewed articles published in Europe only from 1990 to 2020. The results of the two studies do not differ substantially.

The authors correctly highlight the main limitations of the study. Nevertheless, being it a European focused study, and considering the high number of projects funded by the European Union within the LIFE programme, that have included actions to mitigate the impact of large carnivores on livestock, and considering that most projects do not produce peer reviewed publications as their interventions are very localized, the authors risk of missing the majority of information available on this topic, and fail to make a significant contribution to the previously published reviews.

Introduction: L56 – I find the concept of human-wildlife conflict being misused here. The authors may refer to human-wildlife interactions, which might have an impact on human activities that, if not properly addressed and managed, could lead to conflicts among people holding different interests.

Methods: L174 – Not clear why the authors use a 1-5 point Likert scale if they only use three categories afterwards. Please clarify.

L228-230 – The spit of the study period into 2 sets is convenient for the sake of comparison, but the arbitrary split into two periods of equal length, without any reference to particular events makes it difficult to relate any difference detected to explaining variables. Why not then using 10-year periods from 2000 to 2020? What is the rationale behind setting the starting time at 2003?

Figures 3 and 4 – not clear why the grey bars in different groups have different sizes. Please explain

Conclusions: Not clear why the authors address conservationists while suggesting the continuous monitoring of public attitudes towards large carnivores or when predicting that increasing human-carnivores conflicts will have to be dealt with in the future. It is not a duty of conservationists (who are they, anyway?), but rather of management authorities, which should design adequate management strategies for ensuring long term conservation of protected species.

I think the article could benefit from the inclusion of grey literature. I suggest either to make a greater effort in including grey literature or remove the section related to the implementation of damage prevention measures to mitigate carnivore impact on livestock.

Round 2

Reviewer 1 Report

Dear authors,

thank you for your efforts in revising the manuscript.

I have a few comments left:

When I suggested correcting the citation, I meant the one in on L20-26 on the left.

L144-145 is superfluous, as this information can be included on L135: “…scientific research (Tab. 1; Fig. 1).”

L138: move this sentence to L135.

L146: the table capture should be placed above the table. Please check the journal’s guidelines for formatting.

You stated in your previous response nr. 10 that “Any reference to statistical analyses was removed.“, but in the revised manuscript, the text is identical to the first submission.

Even if the focus of the study is on the EU Member State, I strongly recommend including a few sentences discussing the global context. This way the article will be of interest to a wider audience.

Reviewer 2 Report

The authors have addressed nearly all the comments previously made and the manuscript now results much more focused and clear.

The explanations provided are convincing, although some weaknesses remain, and the authors acknowledge them in the conclusion section (e.g., the lack of consideration for gray literature, which I think could provide a significant contribution; or the comparison among periods that have significantly different numbers of records). Notwithstanding the limitations, the manuscript represents a useful contribution, as it provides an overall picture of the various stakeholders studies performed across Europe.

I am not an English native speaker but I had the impression that some sentences could benefit from some shortening. 
